# Systematic Review and Meta-Analysis of Mass Spectrometry Proteomics Applied to Human Peripheral Fluids to Assess Potential Biomarkers of Schizophrenia

**DOI:** 10.3390/ijms23094917

**Published:** 2022-04-28

**Authors:** João E. Rodrigues, Ana Martinho, Catia Santa, Nuno Madeira, Manuel Coroa, Vítor Santos, Maria J. Martins, Carlos N. Pato, Antonio Macedo, Bruno Manadas

**Affiliations:** 1CNC—Center for Neuroscience and Cell Biology, University of Coimbra, 3004-504 Coimbra, Portugal; joao.e.a.rodrigues@gmail.com (J.E.R.); anajmartinho@gmail.com (A.M.); catiajmsanta@gmail.com (C.S.); martins.mjrv@gmail.com (M.J.M.); 2CIBB—Centre for Innovative Biomedicine and Biotechnology, University of Coimbra, 3004-504 Coimbra, Portugal; coroaofc@gmail.com (M.C.); vitorsantos74@gmail.com (V.S.); 3Faculty of Medicine, University of Coimbra, 3004-504 Coimbra, Portugal; nunogmadeira@gmail.com; 4Psychiatry Department, Centro Hospitalar e Universitário de Coimbra, 3004-561 Coimbra, Portugal; 5CIBIT—Coimbra Institute for Biomedical Imaging and Translational Research, University of Coimbra, 3000-548 Coimbra, Portugal; 6Medical Services, University of Coimbra, 3004-517 Coimbra, Portugal; 7Department of Psychiatry and Behavioral Sciences, SUNY Downstate Health Sciences University, Brooklyn, NY 11203, USA; carlos.pato@downstate.edu; 8III Institute for Interdisciplinary Research, University of Coimbra (IIIUC), 3030-789 Coimbra, Portugal

**Keywords:** proteomics, mass spectrometry, schizophrenia, biomarkers, human peripheral fluids

## Abstract

Mass spectrometry (MS)-based techniques can be a powerful tool to identify neuropsychiatric disorder biomarkers, improving prediction and diagnosis ability. Here, we evaluate the efficacy of MS proteomics applied to human peripheral fluids of schizophrenia (SCZ) patients to identify disease biomarkers and relevant networks of biological pathways. Following PRISMA guidelines, a search was performed for studies that used MS proteomics approaches to identify proteomic differences between SCZ patients and healthy control groups (PROSPERO database: CRD42021274183). Nineteen articles fulfilled the inclusion criteria, allowing the identification of 217 differentially expressed proteins. Gene ontology analysis identified lipid metabolism, complement and coagulation cascades, and immune response as the main enriched biological pathways. Meta-analysis results suggest the upregulation of FCN3 and downregulation of APO1, APOA2, APOC1, and APOC3 in SCZ patients. Despite the proven ability of MS proteomics to characterize SCZ, several confounding factors contribute to the heterogeneity of the findings. In the future, we encourage the scientific community to perform studies with more extensive sampling and validation cohorts, integrating omics with bioinformatics tools to provide additional comprehension of differentially expressed proteins. The produced information could harbor potential proteomic biomarkers of SCZ, contributing to individualized prognosis and stratification strategies, besides aiding in the differential diagnosis.

## 1. Introduction

### 1.1. Neuropsychiatric Disorders

Psychiatric disorders (PD) comprise a wide range of mental health problems that can severely impact the well-being of those affected [1,2]. This set of clinical conditions can affect people of all ages and be a leading cause of morbidity, even in childhood and adolescence [3,4]. The effects of PD on public health are profoundly adverse and hugely contribute to the world’s burden of the disease [1,3]. About 10% of the world population is affected, with mental disorders making up 30% of the global burden of non-fatal disease (WHO 2016) overcoming cancer and cardiovascular disease, while 1 million people worldwide die annually from suicide [5].

Thus, the global situation is bleak, with more than 450 million people worldwide living with some form of mental illness; in the European Union only, the number of individuals affected per year is around 165 million [6,7]. Moreover, it is estimated that one-quarter of the world’s population will manifest at least one mental disorder in some period of their life [7,8]. Unfortunately, for several reasons, progress in understanding PD has been slow [1,9].

### 1.2. Schizophrenia

The genetic architecture of schizophrenia is highly complex and heterogeneous. It is characterized by rare mutations that recently emerged with relatively high risk and common variants with individually minor effects on the disease [10]. Genes implicated by both common and rare alleles operate in crucial pathways for brain development, including histone modification, neuronal migration, transcriptional regulation, immune function, and synaptic plasticity [11].

People living with this disease have a significantly reduced average life expectancy, ~20 years lower than the general population. Nonetheless, the mortality rates are high across all age groups [8,12]. The current diagnosis of schizophrenia is mainly based on phenomenological observation and clinical descriptions using the standard operational criteria defined in systematic classifications, namely the Diagnostic and Statistical Manual of Mental Disorders, edition five (DSM-5), and International Classification of Diseases, version 11 (ICD-11), published by the American Psychiatric Association and WHO, respectively [3,13,14]. The main problem is that these diagnostic definitions have relatively good reliability but no established validity [15].

Epidemiologic studies show that it can take up to several years between symptom onset and diagnosis; evidence suggests that the earlier the diagnosis, the better the prognosis, by decreasing the duration of untreated psychosis [16,17].

The symptoms, which typically arise during adolescence or early adulthood, are defined as: (i) positive, such as hallucinations, delusions, and thought disorder; (ii) negative, such as poverty of speech or alogia, lack of motivation and social withdrawal; and (iii) cognitive symptoms, such as attention and learning deficits. While positive symptoms can stabilize throughout the course of illness, negative symptoms tend to increase and become chronic along with cognitive impairments [18,19,20], although currently available interventions, such as antipsychotics and cognitive remediation, can reduce negative and cognitive symptomatology [21,22].

Psychotic symptoms, which integrate positive symptoms, are a defining feature of SCZ spectrum disorders, and their onset defines the first episode of psychosis [23,24]. Despite being considered the main feature for disease onset and diagnostic recognition, psychotic disorders are characterized by an earlier stage, a pre-psychotic stage termed prodrome, which is usually missed by clinicians [25,26].

The treatment of patients is usually based on antipsychotic (AP) medication. After the first successfully employed drug in 1952, chlorpromazine, in the treatment of positive symptoms of SCZ, more drugs were introduced and upgraded in the following years [27,28]. However, they are still ineffective for around 40% of the patients, and some of them end up discontinuing the treatment or having severe side effects [3,29]. The rates of comorbid illnesses associated with SCZ are high, with patients usually linked to an increased metabolic syndrome risk, estimated at 32.5% in SCZ patients in a study by Mitchell et al. 2013 [12]. Metabolic dysfunction is present even in the early phases of SCZ, possibly reflecting specific neuropathological dysfunctions [30].

The pathophysiology of SCZ remains unclear, lacking a comprehensive view of the underlying neurobiological mechanisms, although some aspects are beginning to be clarified. Dopaminergic dysfunction has been one of the pathophysiological hypotheses defended for decades, under various formulations, and is supported by genetic findings [31].

Hypo and hyperactivities of the dopaminergic system are seen in SCZ patients, and both are linked to the symptoms previously described [32,33]. Additionally, other dysfunctions underlying the pathophysiology of SCZ, such as neurotransmitter signaling of glutamate, hypothalamic-pituitary-axonal (HPA) axis signaling, immune system dysregulation and synaptic plasticity anomalies have been reported [19,33,34]. Changes in brain structures, which have also been proposed as etiologically relevant, are correlated with some of these alterations [34].

Despite the efforts to elucidate the mechanisms or etiology behind neuropsychiatric disorders, they remain elusive and not yet clarified. As biomarkers can reflect changes in central nervous system (CNS) diseases, namely the dysregulation of molecular expression profiles, the need to search for reliable biomarkers is becoming imperative, hopefully improving the misdiagnosis of patients [3].

### 1.3. The Search for Biomarkers

To improve knowledge about these complex disorders, “omics” approaches have emerged to shed light on disease pathogenesis and support a trustworthy way of predicting and diagnosing PD [20,35]. With a vast potential associated, high-throughput omics technologies can be a solution to predict clinical endpoints, with the improvement of patient care and outcomes as the ultimate goal. However, the translation from research to a successful clinical omics-based test is far from the great potential of these approaches [36,37].

The search for candidate biomarkers is one of the outputs of -omic studies. According to the National Institute of Health (NIH), a biological marker, generally just termed as a biomarker, is a “characteristic that is objectively measured and evaluated as an indicator of normal biological processes, pathogenic processes, or pharmacologic responses to a therapeutic intervention” [38]. The study of the brain and the associated disorders is complex since it presents a high degree of inter- and intra-cellular heterogeneity; so, different locations may have a distinct proteome due to modifications in different cell types and cellular networks. The CNS proteome can change even with minimal alterations in the normal course of its development and/or function [39,40]. To understand the alterations and the mechanisms related to a disorder, we should analyze qualitative and quantitative changes in the complete set of proteins encoded by an organism’s genome at different or specific points in time [23,41]. Proteomics can be a powerful tool since it can give a real-time evaluation of an individual state, health vs. disease, and, in an ideal scenario, predict the susceptibility to develop a specific mental disorder [4,39]. The possibility of identifying and quantifying the proteins makes the proteomic approach more reliable for evaluating psychiatric diseases at different stages. Moreover, protein-based tests can offer the nearest view of the pathophysiological process behind PD since their expression and function are the results of what happens during post-transcriptional (e.g., alternative mRNA splicing) and post-translational events (e.g., phosphorylation, glycosylation, oxidation), as well as the interactions between them [3,4,42].

The discovery of biomarkers in neuroscience is challenging but may reveal disease-related alterations and, consequently, improve clinical settings; for instance, helping to predict diagnosis, even before the onset, patient stratification, and monitoring disease progression and treatment [35]. Early and guided interventions will improve patients’ outcomes as they are usually prescribed with medication that will not elicit a proper response or even prove ineffective, and it will have to be altered until the desired response is achieved (trial-and-error testing). Moreover, a change in considered diagnoses is also common. Therefore, it would increase the quality of life of individuals and reduce the burden associated with psychiatric disorders, namely misdiagnosis, high rates of hospitalization, and treatment expenses, which have a massive impact on health costs [3,43,44].

### 1.4. Biological Markers in Psychiatric Disorders

The search for biomarkers in psychiatric disorders began with *post-mortem* brain tissue and cerebrospinal fluid (CSF). In contrast to body fluids, brain tissue can only be accessed during autopsies, which is not relevant for disease diagnosis or longitudinal studies. Additionally, some common variables and confounding factors, such as *post-mortem* interval and pH range, can impact this tissue’s integrity. The contribution to protein degradation, as well as medication and age, also leads to drawbacks that cannot be avoided [23,43]. More recently, the whole-body concept emerged since the integration of the brain and various physiological conditions are now known to be reflected in the contents of peripheral body fluids [23,45]. This link between the brain and the periphery enhanced the search for biomarkers in body fluids that could be easily accessible, such as blood [3].

### 1.5. Mass Spectrometry

Since its development, mass spectrometry (MS)-based technologies have been improved and, in recent decades, became a well-suited method for biomarker discovery, supporting the expansion of the proteomics field [46,47]. The success of MS in proteomics is due to its specificity and sensitivity, which are mainly attributable to advances in liquid chromatography coupled to tandem MS (LC-MS/MS) approaches. This type of technology can reveal proteome insights at the composition, structure, and function level. Proteomics tools make it possible to evaluate the proteins in complex biological samples qualitatively and quantitatively (either relative or absolute) [48,49].

In the beginning, successes in proteomics approaches were supported by two-dimensional gel electrophoresis (2-DE), with complex protein mixtures being separated by their molecular charge (isoelectric point) and mass (molecular weight) in the first and second dimensions, respectively. This approach calculates protein abundances based on stained protein spots’ intensities, followed by MS analysis for protein identification [23,50,51]. Although improvements were made, other methodologies emerged to circumvent some of the previous technical drawbacks, namely to face the dynamic range limitations and the unsuitable separation and detection of some protein subtypes, such as membrane proteins [23,48]. Throughout the years, improvements in proteomics approaches were achieved, and a variety of more in-depth MS-based methods were quickly applied to compare protein profiles, usually between control versus disease states. Considering this, there are two main groups within quantitative proteomics methods: (i) labeling techniques, which involve different isotopic labeling of samples, including chemical, enzymatic or metabolic labeling, followed by MS analysis; and (ii) label-free techniques, where the sample is individually analyzed without the addition of any other chemical compound. The newest quantitative approaches are regarded as versatile and cost-effective alternatives to labeled quantitation, having gained significant interest in recent years, mainly due to the development of more sensitive and reliable methods. Additionally, some techniques capable of detecting either relative or absolute peptide levels can provide a targeted MS approach and be used as a validation method [51,52,53].

The absence of molecular biomarkers being used in the clinical environment and the increasing use of large proteomics screenings to search for SCZ biomarkers, allowed us to perform this work, by providing a systematic review and meta-analysis on the use of MS-based methods in proteomic studies to assess biomarkers or a panel of biomarkers associated with SCZ based only on the analysis of peripheral fluids.

## 2. Method

As this study used systematic review and meta-analysis strategies, ethical approval and an informed consent statement were not required. We included the articles that met all keywords that specified the study’s objective. The presented systematic review followed a methodological protocol based on the PRISMA Statement, which was registered in the PROSPERO database, with the identifier: CRD42021274183.

### 2.1. Search Strategy

Research manuscripts included in the systematic review were identified through a computer-based search conducted in two independent databases: PUBMED and Web of Science (WoS). The search was performed in all fields, using the following keywords: SCHIZOPHRENIA AND, PROTEOMIC* AND MASS SPECTROMETRY, until December 2020. Figure 1 presents the flow chart of the selection process of the studies included in this systematic review, following PRISMA 2020 [54]. Searches were restricted by language (English). The PUBMED and WoS databases were last searched on 2 August 2021. Moreover, references in all relevant studies were screened for research papers that might have been missed during the database searches. Two distinct observers, JR and AM, performed the literature search independently to identify articles that potentially met the inclusion criteria, and disagreements were discussed with a third author, BM. Extracted data were entered into a computerized spreadsheet for analysis. Then, the reference lists of the included studies were scrutinized, and excluded studies and previous reviews were searched. The study authors were contacted to request additional information when necessary.

### 2.2. Eligibility Criteria

Studies were included if they met the following criteria: (a) research design included the use of mass spectrometry-based techniques for proteome profiling and/or quantification; (b) studies performed in human peripheral fluids samples, collected with minimally-invasive or non-invasive sampling procedures (which resulted in the exclusion of CSF samples as their collection in many countries is not a standard procedure for psychiatric disorders); (c) research design included a group of identified SCZ patients and a control group comprising healthy controls, and (d) a peer-reviewed English language journal.

### 2.3. Data Extraction

Two authors, JR and AM, independently extracted the following data from the eligible studies, according to a pre-specified protocol of data extraction (Table 1 and Table 2): (1) authors; (2) DOI; (3) year of publication; (4) participants characteristics (including diagnosis type, sample size, and group comparison, mean age, mean illness duration, gender, medication status, type of peripheral samples, and clinical criteria applied); (5) analytical technique; (6) sample preparation (protein depletion or/and enrichment); (7) differences between protein levels of SCZ patients as measured against controls or other mental disorders; and (8) altered pathways.

Any discrepancies between the extracted data were resolved in a group meeting.

### 2.4. Quality of Evidence

The quality of the studies was determined using the QUADOMICS methodology criteria (Appendix A), and it was evaluated independently by two authors (Appendix A). QUADOMICS is an adaptation of QUADAS—a quality evaluation tool for systematic reviews of diagnostic accuracy studies, accounting for technical particularities presented by omics methodologies [55,56].

### 2.5. Statistical Analysis and Gene Ontology Analysis

To perform the meta-analysis, the effect size for each measured protein was standardized to log2 FoldChange. In this way, effect sizes and corresponding significance that were heterogeneously expressed in the studies such as: (i) ratio or log(ratio) and the corresponding *p*-value; or (ii) group averages and the corresponding standard deviations were all transformed into fold change and corresponding *p*-values. Proteins in which it was possible to compute the effect size in at least two research studies were included in the meta-analysis.

A Forest Plot was created to present the output data, being the conventional way to report meta-analysis results. Meta-analysis was performed in R version 4.0.3 combined with Rstudio, using the following R packages: ‘meta’ [57], ‘metafor’ [58], and ‘dmetar’ [59]. Gene ontology analysis was performed using MetaboAnalyst 5.0 [60] and the KEGG Mapper Color tool [61,62].

## 3. Results

### 3.1. Characteristics of Included Articles

The search strategy followed for selecting the eligible studies included in our systematic review/meta-analysis is shown in Figure 1. From the searches performed in WOS and PUBMED databases, a total of 313 potentially relevant research manuscripts were identified in the initial screening. No additional studies based on a manual search were identified for inclusion. Based on the abstracts’ review, 191 were retrieved for more detailed evaluations. Of these research manuscripts, 171 were excluded after full-text reading (abstract reading was not enough to exclude these articles immediately, and despite the match of keywords, the studies did not fit in the inclusion criteria), with 46 articles identified as reviews, 44 studies were performed in mice/rats, 15 studies were conducted in cell lines, 46 studies analyzed brain tissue, 10 reports with samples collected with invasive sampling procedures, seven reports had no healthy control group and one study with no clear SCZ patients (the identified disease was described as a psychotic episode). Additionally, three articles were excluded as they did not have any information related to the proteome profile. In total, 19 studies met all eligibility criteria and were included in this systematic review.

The essential characteristics of the 19 eligible studies are shown in Table 1. As mentioned, only studies of human peripheral fluids were considered.

All articles included in this review were published from 2007 and onward; of those, 17 after 2010, reflecting the recent interest in MS proteomics strategies for biomarkers exploration in SCZ and its biological pathways. More detailed information about the studies is summarized in Table 2 with the indication of cohort information, biological sample and type of sampling, diagnostic criteria, treatment information (treated or drug naïve), type of MS-based method, other techniques applied, use of depletion, and/or enrichment approaches, differentially expressed regulated proteins identified, altered pathways and significant findings.

### 3.2. Cohort Information

All selected studies (*n* = 19) had a clinical control group, comparing a group of individuals with SCZ and a control group to identify diagnostic biomarkers. Within these, one interesting study used a control group that distinguishes smokers and non-smokers from healthy individuals [74]. Although not included in this review due to the lack of a healthy control group, two other studies assessing the effects of pharmaceuticals in SCZ therapeutics [82,83] will be highlighted later. Additionally, despite fulfilling most inclusion criteria, three studies did not have any protein profile information [84,85,86]. In that way, they were not considered for this systematic review.

### 3.3. Number of Samples

The number of patients with SCZ included in the studies varied between 8 [77] to 60 [66,67]. Compared with the first studies, the last two years of publications show an increasing trend in the number of patients per study (see Figure 2). The two studies with the higher cohort of individuals with SCZ were recently published, 2017 [67] and 2019 [66], and both with around 60 SCZ patients.

Of the four works published in 2019 and onward, three studied more than 30 individuals with SCZ [63,64,66]. On the other hand, in 2017, only one of five published studies had a cohort of SCZ composed of more than 30 individuals [67]. In accordance, until 2012, only one of seven studies used a number of individuals with SCZ higher than 30 [81], interestingly being the first article using MS proteomics strategies in peripheral fluids to assess SCZ. Since 2015, eleven studies have been published, and half of them studied more than 30 individuals with SCZ, whereas before 2014, only two of the eight articles published had more than 30 SCZ patients. This information clearly shows that more recent studies privilege larger cohorts, a reliable parameter to achieve more significant results.

### 3.4. Diagnosis Criteria

It stands out that DSM-IV was the most used diagnostic criteria (11 studies), while one of the studies also used ICD (see Table 1) [74]. An older and the most recent DSM manual available—DSM-III and DSM-V, respectively—were applied in one study each; DSM-III was used in one study published in 2007 [81], whereas the recent DSM-V was applied in one study published in 2019 [64]. ICD-10 was the second most used criteria (seven studies), applied together with ICD-9 in one study [67].

### 3.5. Age

Throughout the studies, the reported the average age of the studied individuals was comprised between 16 [77] and 52 [75] years, being the first study the only one reporting an average age below 25 years [77]. The majority of the studies have an average age between 29 and 43 years (16 studies) for the SCZ group. Only one study lacks information about the age of the individuals [74]. In concordance with the information provided by the studies that usually compare between age-matched groups, the average age was similar in the different groups in the studies.

### 3.6. Gender

Considering the gender information in the SCZ group (see Table 1) provided for 18 of the 19 studies in this analysis, only three studies used a proportional number of samples per gender [67,75,76]. In 13 of the 18 studies, male patients prevailed, in some cases with three times more samples than the female gender [66,69,73]. Only three studies had more samples from the female gender [63,68,72], with two of them having minimal differences between genders [68,72].

In the control group, the differences in gender are not so exacerbated, with a similar number of studies having more male (*n* = 8) than female (*n* = 7) samples. Once again, three articles used a similar number of samples representing each gender [75,76,81], but only two of them match the number of samples and gender used in the SCZ group [75,76]. Only two studies that used a control group did not provide gender information [74,77].

Considering the five studies with a group of other disorders (BD, depression, and other disorders), two studies had a higher number of males [63,69], and three had a higher number of females [65,70,72]. Only two followed the same gender ratio in the SCZ and control groups [69,72].

### 3.7. Illness Duration

Only a few studies referred to illness duration (*n* = 6), with a period comprised between seven and twelve years [63,64,65,69,70,73]. Of the six studies that contain this information, three articles were published in 2019 [63,64,65], two in 2017 [69,70], and one in 2015 [73], with no article before 2014 containing data about illness duration. This reflects an increased awareness in the last years of the importance of this information in assessing and evaluating the disease. Although it is more common to see this information reported in studies published in the last five years, there is a clear need to standardize the type of data reported in these studies.

### 3.8. Frequency of the Publication

It can be observed that the publication of studies on this subject does not follow a pattern throughout the last fifteen years (Figure 2). More precisely, in 2009, 2013, 2016, and 2018, we could not find any articles that fit our research strategy. However, it is noticeable that there has been an increase in publications over the years.

### 3.9. Type of Sample and Sampling

Based on the studies retrieved from the database search (Figure 1), it is possible to notice that brain tissue is still being analyzed, but the number of studies with body fluids is increasing (see Figure 3), providing some evidence that it can be a reliable choice for biomarkers research [87]. Biofluids are suitable matrices that enable more user-friendly tests, as the majority of them are easy to access [88,89]. Based on their accessibility, they can be categorized as non-invasive (saliva, sweat, urine, and tears), minimally invasive (blood), and invasive (cerebrospinal fluid).

The most prevalent biological sample type studied was serum in 10 studies, followed by plasma in 4 studies [64,66,69,81] and PBMCS in 3 studies [68,78,80] (Table 2 and Figure 3). Only one study was performed using saliva [74] and another using sweat [77]. Since 2015, studies conducted in serum samples have been dominant, with serum being analyzed in seven studies, although plasma samples were analyzed in three studies [64,66,69]. During this period, only one study used PBMCs [68]. Before 2015, a substantial heterogeneity of sample types was observed; from the total of eight articles published during this period, three studies analyzed serum [75,76,79], two studies used PBMCs [78,80], and one study used plasma [81], saliva [74] and sweat [77].

As observed, plasma and serum are the main sample types used in the studies (Table 2 and Figure 3). These biofluids can be easily sampled and have been widely used in proteomics-based research and disease diagnosis. However, the complexity and dynamic range that characterize the proteome of both samples are responsible for unsatisfactory outcomes in the search for disease biomarkers. As potential biomarkers are usually present in low concentrations and tend to be masked by highly abundant proteins, some strategies have been applied to overcome this challenge. Depleting high-abundance proteins and enrichment of low and medium abundance proteins are two of the most used methods to circumvent this problem [90]. Considering the selected studies (Table 2), there has been consistent use of depletion strategies throughout time, with four studies [63,66,70,71] using depletion in the last five years and three studies in the previous period [75,76,79]. Similar to depletion, enrichment strategies were used in seven studies, four after 2015 [64,66,72,73] and three before 2012 [76,78,80]. From a total of seven studies that used enrichment techniques (proteominer, aptamers, IMAC, IMAC30, C18, TiOtips, and subcellular fractionation), only two were not analyzing serum or plasma but PBMCs [78,80]. Among the ten studies that analyzed serum samples, only two [65,67] did not use depletion/enrichment techniques, whereas, from the four studies examining plasma, two [69,81] did not use those approaches.

A clear trend of individual samples analysis was observed in 15 out of 19 studies. Only four studies worked with pooled samples, with three of them published after 2017 [65,66,70]. The other study, analyzing sweat samples, was published in 2012 [77].

### 3.10. Drug Naïve or Minimally Medicated

Regarding patients with and without treatment (drug naïve/minimally medicated), it is noticed that there is a prevalence of studies with treated patients, 14 out of 19 studies. Five studies were performed in drug naïve/minimally medicated patients [63,67,75,76,80], two after 2017, whereas the other three articles were published before 2013.

Additionally, two studies worked exclusively with a single cohort of individuals with SCZ, which were analyzed before and after treatment [82,83]. These were not included in the systematic review since they did not have a healthy control group; however, they were included in the discussion.

### 3.11. MS-Based Methods

Overall, it is undeniable that LC-MS/MS analysis was the prevalent MS-based technique among the studies considered in this review, with 13 out of the 19 studies. The other six studies applied MALDI TOF/TOF or SELDI TOF/TOF analysis, two after 2017 [68,71], two in 2015 [72,73], and the remaining two studies before 2012 [80,81]. In recent years, an increase in the application of in-gel digestion followed by LC-MS/MS analysis has been observed, as we can see from three (out of four) articles published in 2019 [63,64,66]. Additionally, one study also applied ICP-MS-based methods to assess the interactions between metals and proteins [65].

### 3.12. Validation and Other Techniques

Other techniques have also been applied to validate results or further characterize the cohorts with two main strategies: untargeted and targeted approaches.

For targeted/validation approaches, immunoassay methods have been applied when validating specific proteins identified as differentially expressed. Around half of the studies (9 out of 19 studies) used immunoassays methods for validation of protein’s expression pattern, with the enzyme-linked immunosorbent assay (ELISA) being the most used approach [63,66,75,76,79,82], followed by Western blot (WB) [64,78].

For untargeted approaches, one study applied magnetic resonance imaging (MRI) to detect underlying morphological changes occurring in SCZ and BD patients [69]. GC-MS and FTIR techniques were also used in one study [73] for a multi-platform metabolome and proteome profiling study to identify a prospective biomarker for SCZ.

## 4. Main Studies Performed

### 4.1. Schizophrenia vs. Healthy Control (n = 19)

The comparison between SCZ and a healthy control group was the primary purpose of this systematic review expressing the main objective of establishing a proteomic profile characteristic of the disorder and identifying specific proteins that could be used in the future to help SCZ diagnosis.

Considering studies that analyze blood-related samples (plasma, serum, and PBMCs), a total of 197 proteins were identified as differentially expressed between SCZ patients and healthy controls. The higher number of proteins identified as altered were found in studies that analyzed serum samples (total of 131 proteins), reflecting that serum was the most analyzed biological fluid, followed by plasma (total of 66 proteins) and 21 proteins in PBMCs. No protein was identified as altered in all three blood-related matrices; 20 proteins were coincident between serum and plasma, namely: the apolipoproteins A1 (P02647), A2 (P02652), A4 (P06727), C1 (P02654), C2 (P02655), C3 (P02656), D (P05090), E (P02649), and F (Q13790), alpha2-antitrypsin (P01009), alpha-2-antiplasmin (P08697), antithrombin-III (P01008), complement factor B (P00751), clusterin (P10909), complement C4-A (P0C0L4), ficolin-3 (O75636), coagulation factor XIII B chain (P05160), haptoglobin (P00738), retinol-binding protein 4 (P02753), and transthyretin (P02766); and one protein was identified in both plasma and PBMCs: alpha defensin 1 (P59665). The summary of the number of proteins identified as altered is shown in Appendix A (see also Appendix A).

Moreover, there are proteins identified in other human peripheral fluids, namely in sweat (total of 19 proteins) and saliva (total of 8 proteins) (see Appendix A). Considering all peripheral fluids, a total of 217 proteins were identified as altered. Overall, one protein was identified in both PBMCs, plasma, and saliva (alpha defensin 1, P59665); two proteins were identified between PBMCs and saliva (alpha defensin 2 and 3, accession numbers: DEF2 and P59666, respectively); and one protein coincident in several pair comparisons, namely between serum and sweat (dermcidin, P81605); plasma and sweat (parkinson disease protein 7, Q99497); PBMCs and sweat (glyceraldehyde-3-phosphate dehydrogenase, P04406); and saliva and sweat (cystatin A, P01040). The summary of the number of proteins identified as altered in all human peripheral fluids is shown in Figure 4.

It should be noticed that for alpha defensin 2 protein (DEF2), found in PBMCs and saliva, no information about protein ID was found; we could not find the corresponding accession number/identifier through the UniProt database [68,74]. Moreover, in a serum study, one protein (Immunoglobulin kappa variable 3–20) had its accession number updated, considering UniProt data (from P04206 to P01619) [75].

### 4.2. Schizophrenia vs. Bipolar Disorder (n = 5)

In total, five studies had a comparison between SCZ and BD, with higher prevalence after 2017 (*n* = 4) [63,65,69,70], against one study before that date [74]. The prevalence in recent years reflects an increasing interest in identifying differentially expressed proteins between these two major mental disorders, seeking the definition of disorder-specific biomarkers to help with diagnostic specificity.

The four BD vs. SCZ studies published after 2017 were all performed in blood-related samples, with three studies in serum [63,65,70] allowing the identification of 48 altered proteins; and one study in plasma [69] identifying 25 altered proteins (see Appendix A). One of the studies using serum samples performed the comparison of the ionomic profile of SCZ and BD disorders against a control group to establish relationships between metals and proteins [65]. Overall, 70 differentially expressed proteins between BD and SCZ patients were identified in blood-based matrices. By assessing these proteins, it was observed that three proteins were coincident in plasma and serum samples: apolipoprotein D, apolipoprotein E, and retinol-binding protein 4 (P05090, P02649, and P02753, respectively). Interestingly, these three proteins were also highlighted in SCZ vs. control studies, altered both in plasma and serum samples.

In the study using saliva samples, eight proteins (α-defensins 1 to 4, S100A12, cystatin A and S-derivatives of cystatin B, cystatin B S-glutathionyl, and cystatin B S-cysteinyl) were identified as altered between BD and SCZ against control; however, no statistically significant differences were observed between the SCZ and the BD groups [74].

Only one study compared the proteomic profile of SCZ and other disorders, namely depression [72]. The study was performed in serum and allowed the identification of one altered protein corresponding to N-terminal fragments of fibrinogen.

### 4.3. Drug Naive vs. Treated (n = 2)

Although not selected for this systematic review since they did not have a healthy control group in the study, two studies compared patients with SCZ before and after treatment [82,83]. In one of these studies [82], the patients were treated for 8 weeks with the AP risperidone, and the analysis was performed before and at the end of this treatment. All patients included did not undergo prior AP treatment. This study aimed to evaluate the change in plasma protein expression levels and elucidate potential biomarkers related to metabolic side effects as a consequence of risperidone treatment, allowing the identification of 18 proteins up or downregulated after the 8 weeks of treatment. In the other study, a comparison between a group of SCZ drug naïve/minimally medicated and the same group after treatment was performed [83], with the patients under different AP therapy: (i) olanzapine (*n* = 18); (ii) quetiapine (*n* = 14) and (iii) risperidone (*n* = 26). The study aimed to unravel molecular pathways implicated in the efficacy of drug response, allowing the comparison between patients who responded or did not respond to treatment. In total, 23 identified proteins were shared between both groups (with 13 proteins following the same behavior trend). 

### 4.4. Bias Analysis

The results of evaluating the quality of the proteomic studies included in the systematic review are displayed in Appendix A. The least fulfilled QUADOMICS quality criteria were items 4—factors influencing sample collection (13 studies); 11—reference standard description (13 studies); and 16—prevention of overfitting (14 studies). The majority of the studies did not report enough data to assess items 6, 7, 9 and 12—respectively, the time period between the reference standard and index test, reference standard’s ability to correctly classify target condition, consistency of reference standard use despite index test’s results, and blind interpretation of reference standard.

### 4.5. Meta-Analysis

Ten out of the nineteen [64,65,67,69,71,75,76,78,79,81] included in the systematic review reported data in a format amenable to meta-analysis, providing data about the differently expressed proteins between SCZ and control groups in the form of effect size (average or fold change) and error deviation (standard deviation or *p*-value).

It was possible to compute the effect size (expressed in log2 Fold Change) of proteins if identified in at least three independent studies. In total, the meta-analysis between schizophrenia patients and healthy controls was performed for six proteins: apolipoprotein A1 (APOA1, P02647), apolipoprotein A2 (APOA2, P02652), apolipoprotein A4 (APOA4, P06727), apolipoprotein C1 (APOC1, P02654), apolipoprotein C3 (APOC3, P02656), and ficolin-3 (FCN3, O75636) (genes names and accession numbers, respectively). A meta-analysis included these proteins to assess their overall expression change. The forest plot (95% CI, confidence intervals) is shown in Figure 5.

The meta-analysis results suggest several apolipoproteins as potential disease biomarkers, with APOA1, APOA2, APOC1, and APOC3 showing a decreased tendency in SCZ patients compared with healthy control subjects. Heterogeneity was significantly observed for APOA2 and APOC3 (*p* < 0.01, *I*^2^ > 80%), and for APOA1 and APOC1 (*p* = 0.01, *I*^2^ = 78%). The protein ficolin-3 (FCN3) also showed a consistent trend, with all the three studies identifying FCN3 as up-regulated in SCZ patients (*p* < 0.01, *I*^2^ = 87%). For APOA4, the results found were not consistent, being identified as downregulated in three studies and upregulated in one study.

## 5. Discussion

In this work, a comprehensive systematic review was performed along with a meta-analysis to evaluate mass spectrometry-based proteomics applied to human peripheral fluids to assess biomarkers of schizophrenia and the identification of relevant networks of biological pathways. 

The major studied topic was the assessment of protein expression differences between SCZ patients and healthy controls (CTR). A total of 217 proteins were identified as altered between SCZ and healthy control groups in peripheral fluids, including serum, plasma, PBMCs, sweat, and saliva.

Apolipoproteins (APOs) were the group of proteins mostly reported in SCZ vs. control studies as differentially expressed. In fact, ten studies reported the dysregulation of apolipoproteins [63,65,66,67,69,71,75,76,79,81]. APOs are very important in lipid homeostasis by transporting cholesterol and lipids between cells, having a well-established role in the transport and metabolism of lipids, and in inflammatory and immune response regulation [91,92]. This group of compounds has been indicated as potential candidates for psychiatric biomarkers, with several studies reporting altered levels of cholesterol and APOs in psychiatric disorders [92,93,94]. Accordingly, in the selected studies, APOs alterations were associated with inflammatory response [67,79,81], immune system [63,76], lipid metabolism [67,76], cardiovascular system [66], retinoid transport [81], and cognitive decline and underlying morphological changes [69]. Several apolipoproteins were identified as altered in the selected studies. APOA1 [69,76,79,81], APOA2 [67,69,76,79], and APOA4 [67,69,71,79] were found as differentially expressed in four studies; APOC1 [67,69,79], APOC2 [69,71,75], APOC3 [67,69,71], APOD [63,69,79], and APOE [63,69,81] were identified in three studies; APOB [66,69], APOF [69,75], APOH [65,67], and APOL1 [69,75] were found in two studies; and APOC4 [69] and APOM [63] in only one study.

APOA1 is the major protein component of the HDL fraction in plasma. Together with APOA2, APOA4, APOC1, and APOD, APOA1 is recognized for regulating the plasma levels of free fatty acids, having an important role in HDL and triglyceride-rich lipoprotein metabolism in the reverse cholesterol transport pathway [95]. APOA1 is also reported as having pro-immune and anti-inflammatory potential [91]. In all selected studies where it was identified as altered, ApoA1 level was reported to be reduced in schizophrenia patients compared to healthy subjects [69,76,79,81].

APOA2, the second most abundant protein in HDL fraction, is a key regulator of HDL metabolism [95], although its inflammation role is not clearly defined, with different studies reporting it as having pro- and anti-inflammatory effects [96]. APOA2 was identified as differentially expressed in four studies, being downregulated in SCZ patients in all studies [67,69,76,79].

APOA4, a lipid-binding protein, is known to be involved in a broad spectrum of biological processes, including lipid metabolism, reverse cholesterol transport, atherosclerosis protection, and glucose hemostasis [97]. APOA4 was identified as differentially expressed in four studies; however, it showed a heterogeneous behavior: downregulated in three studies [67,69,79] and upregulated in only one study [71].

The apolipoproteins APOC1, APOC2, APOC3, APOD, and APOE were identified in three studies as differently expressed, showing a general tendency of downregulation in SCZ patients except for APOE, which has a trend for upregulation. Of these, only for APOD, a soluble carrier protein of lipophilic molecules that is mostly expressed in neurons and glial cells within the central and peripheral nervous system [98], the results were consistent in all three studies, and it was identified as decreased in SCZ patients [58,65,72]. A trend of downregulated behavior was identified for APOC1 (the smallest of all APOs, participating in lipid transport and metabolism) [67,79], APOC2 (a small exchangeable apolipoprotein found on triglyceride-rich lipoprotein particles) [71,79], and APOC3 (an APO capable of inhibiting lipoprotein lipase and hepatic lipase) [67,71], in two out of three studies. Interestingly, the same study reported upregulated levels of these three APOs in SCZ [69], showing contradictory results compared with the other studies. On the other hand, APOE, a protein with a critical function in lipoprotein-mediated lipid transport, was identified as upregulated in two out of the three studies [63,69], reflecting a tendency to be increased in SCZ.

APOB, APOF, APOH, and APOL1 were identified in two studies. APOF [69,75], APOH [65,67], and APOL1 [69,75] had a similar behavior: upregulated in the two studies. For APOB, no clear trend was observed, with one study reporting its increase [66] and another a decrease [69] in SCZ patients.

A set of four proteins were also identified in three or more selected studies as differentially expressed, namely retinol-binding protein 4, RET4 [63,69,76,81], haptoglobin, HPT [67,71,81], antithrombin-III, ANT3 [67,69,81] and ficolin-3, FCN3 [67,69,76]. 

RET4 is mainly expressed in the liver with a primary function to transport retinol (vitamin A) from the liver to peripheral tissues, with retinol being essential for the brain to facilitate learning, memory, and cognition [99]. Retinoid signaling plays a vital role in immune cell function. Accordingly, it is suggested that factors that affect this system could have important implications for SCZ and other psychiatric disorders-associated inflammatory stress [100]. RET was increased in two studies [69,81], whereas in the other two studies, it was decreased [63,76]. Although four studies identified RET4 as altered, this protein was not included in the meta-analysis since the information required to compute the effect size and corresponding significance was unavailable in two studies.

HPT, a positive acute-phase protein that binds free hemoglobin and removes it from the circulation to prevent kidney injury and iron loss following hemolysis, was identified as upregulated in the three studies [67,71,81]. This is consistent with previous studies reporting that SCZ is accompanied by an activation of the inflammatory response system with signs of an acute phase response, such as increased plasma HPT concentration [101].

ANT3, a glycoprotein anticoagulant mainly produced in the liver that exerts anticoagulant and anti-inflammatory effects by targeting activated thrombin and other blood coagulation factors [102], was identified as increased in SCZ patients [67,69,81]. 

FCN3 is a ficolin, a protein containing both a collagen-like domain and a fibrinogen-like domain with a specific binding affinity for N-acetylglucosamine. FCN3 can complex with mannose-associated serine proteases to activate the complement pathway [103], being ficolins’ activation already reported as a potential biomarker of the severity of schizophrenia [104]. In the selected studies, FCN3 was also identified in three studies as upregulated FC [67,69,76].

Although the above-mentioned proteins were identified as differentially expressed in more than two studies, due to the lack of complete statistical information on a format amenable to a meta-analysis in some selected studies (five out of 19), only FCN3 was added to the list of proteins characterized in the meta-analysis.

The α-defensins (DEF1, DEF2, DEF3, and DEF4) were also reported in three studies [68,74,80], all identified as upregulated. Defensins are small cationic peptides with anti-bacterial, antiviral, and immunomodulatory properties [74], divided into three subfamilies (α, β, and θ-defensins) according to the connectivity of three intramolecular disulfide bonds generated by six conserved cysteine residues. The α-defensins 1–4, firstly found to be produced by neutrophils, have been recognized as the secretion products of a variety of leukocytes, including monocytes, B cells, αβ and γδ T cells, and natural killer cells [105]. Defensins have been found to permeabilize cell membranes, act as opsonins targeting microbes for phagocytosis, inhibit protein kinase C, bind to ACTH receptors to block steroidogenesis, and act as chemoattractants for monocytes. DEF1 and DEF 3 are the products of DEFA1 and DEFA3 genes and differ only at the N-terminal residue (Ala in HNP-1 and Asp in HPN-3); DEF2 can originate both from HNP-1 and HPN-3 by enzymatic loss of the N-terminal amino acid residue. DEF4 is expressed from the DEFA4 gene; however, this gene is not duplicated, and only two copies exist in a diploid genome, leading to a noticeably lower amount of DEF4 compared with the other α-defensins [74].

The immune system and inflammatory response were the most identified biological processes altered in SCZ patients [63,67,68,74,76,79,80]. These results agree with current knowledge about SCZ, associating the immune system and inflammatory response with the SCZ pathophysiology [106,107,108]. In fact, a wide range of immune alterations has been reported in SCZ patients, such as elevated levels of cytokines and inflammation markers, abnormalities of the blood-brain barrier, CNS inflammation, and increased autoantibody reactivity [107].

Several other mechanisms have also been linked to SCZ, including mitochondrial dysfunction, energy metabolism processes, complement and coagulation cascades, oxidative stress, transport, morphological changes, cognitive impairment, lipid metabolism, and hypothalamic–pituitary–adrenal (HPA) axis over-activation [87,108]

To validate these findings and integrate the biological meaning of the results from the selected studies, a gene ontology (GO) analysis was performed where all proteins found to be differentially expressed in any of these studies were used (Figure 6).

From this ontological analysis, it is possible to observe that the proteins found as altered in the studies belong to different biological functions; still, the ontologies with the highest impact and significant enrichment (Figure 6A and Appendix A) are cholesterol metabolism (Figure 6B), and other metabolic pathways; complement and coagulation cascades (Figure 6C); and also immune response-related pathways here represented by terms relating to infection and immune system disorders. As discussed above, these are the most prominent pathways discussed in the various reports analyzed, especially because some of the proteins are reported as altered in more than one study (as it is visually highlighted for the two chosen pathways in Figure 6), and even the uniquely reported proteins of each study belong, in general, to some of these pathways.

This general overview of the biological functions of all the proteins that have been reported as altered in SCZ (irrespective of study or peripheral fluid) strongly converge to pathways like lipid metabolism, coagulation cascade, and immune response, suggesting that key players in the (dys)regulation of these pathways may be helpful in the future as biomarkers.

Increased concern on defining disorder-specific biomarkers and understanding associated altered biological pathways led to a recent interest in studying metabolic differences between SCZ and other psychiatric disorders. Five studies (four studies after 2017) performed a comparison between SCZ and BD proteomic profiles [63,65,69,70,74]. Following the findings observed in the SCZ vs. CTR studies, the immune system and inflammatory response were also the most identified biological pathways altered [63,70,74].

Iavarone et al. [74] confirmed a schizophrenia-associated dysregulation of the immune pathway of peripheral white blood cells, suggesting that the dysregulation in BD patients could involve the activation of more specific cell types than that of SCZ.

In de Jesus et al. [70], three unique proteins, namely complement C4-A, complement C4-B, serum amyloid P-component (CO4A, CO4-B, and SAMP, respectively) were identified as differentially abundant between SCZ and BD (higher levels in SCZ), being associated with the inflammatory response.

In Smirnova et al. [63], the definition of the proteome profiles of different groups revealed 27 proteins specific for schizophrenia (not present in BD) and 18 for BD. The particular proteins of schizophrenia mainly were associated with immune response, cell communication, cell growth and maintenance, protein metabolism, and regulation of nucleic acid metabolism, while BD specific proteins were mostly related to immune response, regulating transport processes across the cell membrane and cell communication, development of neurons and oligodendrocytes and cell growth.

In Knochel et al. [69], protein expression in SCZ and BD patients was associated with cognitive deficits and underlying brain structures. The results suggested that detecting molecular patterns in association with cognitive performance and its underlying brain morphology was important to better understand the pathological mechanisms of psychiatric disorders SCZ and BD and, consequently, to support the diagnosis and treatment of both disorders.

In Pessoa et al. [65], a metalloproteomics study was performed, allowing the identification of the proteins IGHG1 (both SCZ and BD), Ig lambda chain V-IV region Hil and ApoH (only in SCZ), and IGKV2D-28 (only in BD) as altered in SCZ and BD comparing to a healthy group and the identification of different concentrations of Li, Mg, Mn, and Zn in BD patients and high levels of Cu for SCZ patients, indicating an imbalance in the homeostasis of essential micronutrients.

Only one study compared SCZ and depression patients [72]. In this study, N-terminal fragments of fibrinogen protein were identified as downregulated in SCZ patients.

## 6. Strengths and Limitations

Mass spectrometry (MS)-based proteomics strategies have a (semi-)quantitative character that allows the generation of different levels of information about the individual proteome, which can lead to a more comprehensive characterization and understanding of the functional alterations associated with the stimulus. The advances occurring in recent times in MS instrumentation (improvements in the sensitivity, accuracy, resolution, and scan speed), combined with more sensitive and selective sample preparation methodologies and with massive advances in the computing capacity (leading to an increase in the data processing and data mining ability), allow MS proteomics strategies to produce a large amount of biochemical information. Taking advantage of the recent computational advances, the application of novel data analysis strategies (e.g., statistical and machine learning approaches, and gene ontology analysis, among others) increases the ability to extract meaningful information of MS data, facilitating the comprehensive mechanistic understanding of the biological processes associated with the disorder.

The study of human peripheral fluids, discussed in this systematic review, represents an attractive approach from a clinical point of view. Peripheral fluids contain disease-associated proteins secreted or leaked from pathological tissues across the body, easily obtained through non-invasive procedures that allow large sample volume collection [109]. Due to its proximity to the brain, cerebrospinal fluid (CSF) is considered relevant when studying brain disorders. Despite being a dynamic fluid, CSF has to be collected through a lumbar puncture, which is an invasive procedure and leads to a minimal amount of fluid, limiting the possibility of applying this type of analysis in CSF samples [23,35]. This shows the importance of selecting more readily accessible samples [23]. Studies with plasma and serum samples have increased over the last few years, and looking at psychiatric disorders as whole-body diseases, somehow contributed to this change [89]. Besides protein content being significantly more abundant than in CSF, approximately 500 mL of CSF are exchanged daily in circulating blood [46,47]. Additionally, dynamic changes can also be studied in serum and plasma samples, which can be collected in reasonable amounts and by straightforward and safe procedures [35].

The main limitations identified in these selected studies are related to the vast heterogeneity observed in the cohorts’ characteristics. Significant differences related to diagnostic criteria used, medical and psychiatric comorbidities reported, psychopharmacological treatment and its duration, as well as the sociodemographic variables such as age, illness duration, gender balance, diet, and use of tobacco, alcohol, or other psychoactive substances, among others, were observed in the selected studies. These limitations were also observed in a BD systematic review [110], suggesting common limiting factors across the psychiatric disorders studies using MS proteomics approaches, thus highlighting the need for more standardized, or at least more controlled, cohort characteristics.

The majority of the selected studies were used as exclusion criteria in SCZ patients with other medical conditions, mainly inflammatory, autoimmune, hepatic, cardiovascular, endocrinological, and metabolic diseases (e.g., cancer, AIDS, diabetes, heart disease) and substance abuse. The exclusion of SCZ patients with current or past psychiatric disorders or family history of other psychiatric disorders was also reported. Only one study mentioned pregnancy and breastfeeding as exclusion criteria. Overall, there are huge discrepancies when detailing exclusion criteria; while there is a clear definition of the diseases that lead to a patient’s exclusion in some studies, there is only a subjective description in many others. Standardization of the exclusion criteria and the sociodemographic conditions is essential, seeking to minimize the confounding factors that may hinder the identification of potential proteomic changes specific to the disorder and the use of data in further meta-analysis.

The information describing SCZ patients’ psychopharmacological treatment and its duration is also highly heterogeneous, adding confounding variables to the analysis. In the selected studies, the characteristics reported varied from drug-naïve/ minimally medicated (not receiving psychopharmacological treatment for at least one week to six months prior to sample collection) to medicated. Studies with medicated SCZ patients are the most common (14 out of 19 studies), whereas drug naïve/minimally medicated conditions were used in five studies. The use of treated vs. drug naïve/minimally medicated conditions is mainly dependent on the research objectives. The use of drug naïve/minimally medicated subjects, minimizing potential confounding factors, is important for a more comprehensive understanding of the pathophysiology of BD and the identification of potential biomarkers of the disorder. However, studies related to the effects of psychiatric drugs to treat BD and the definition of differences in the proteomic profile of patients with BD with those drugs are also essential. Interestingly, the last five years showed no apparent increase in drug naïve/minimally medicated SCZ studies. 

In the selected studies, clear care in using age-matched SCZ and healthy control groups was identified. The concern to use similar average age between groups and share the age characteristics of the groups reveals a research awareness in need of homogeneity of the age parameter between groups.

SCZ cohorts’ average illness duration varied between 7 and 12 years; however, this is not a customarily reported characteristic in SCZ studies (6 out of 19). In fact, of the six studies reporting the illness duration, four of them are studies that also compared SCZ and BD groups. The lack of interest in the illness duration is very different from what is observed in BD studies [110], where this parameter is reported in the majority of the studies.

For gender comparison, the selected studies clearly showed a male gender prevalence, with 13 out of 18 studies having more male patients (in some cases by 3-fold). The authors seemed to be aware of the importance of this information, with only one study lacking gender information since there are indications of gender differences in SCZ. In fact, consistent findings have been reported so far for differences between males and females in the age of onset, premorbid functioning, negative and affective symptoms, and substance use [111,112]. On the other hand, other psychopathological domains apart from negative and affective symptoms, neurocognition, social cognition, and personal resources have received scarce attention, and/or relevant studies provided discrepant findings, not allowing for confirmation of gender differences in these domains of SCZ disease [113,114]. In this way, the use of gender-balanced groups is dependent on the desired outcome. The use of gender-balanced groups is important to minimize confounding factors associated with gender. Since different genders present distinct SCZ characteristics, gender studies are also important to better understand the pathophysiology of SCZ, although BD and control groups must have a balanced number of samples.

The strength of studying drug naïve/minimally medicated SCZ patients, with no other medical and psychiatric comorbidities and clearly defined sociodemographic variables is that the confounding effects of medication and other diseases on the patient’s overall proteome are minimized, allowing a more precise definition of the differentially expressed proteins related with the disorder. Accordingly, the use of standardized sociodemographic, clinical and cognitive variables across the studied groups would lead to more objective and specific studies allowing a more comprehensive understanding of SCZ pathophysiology and, consequently, increasing the possibility of identifying specific biomarkers of SCZ. In addition, studies focused on gender discrepancies are also needed to assess and describe better the importance of gender in the pathophysiology of SCZ. The study of medicated SCZ groups with the same standardized conditions will also be important to evaluate and define more accurate psychopharmacological treatment specific to each individual with specific characteristics.

## 7. Directions for Future Research

The recent advances in MS proteomics strategies applied to human peripheral fluids allow the establishment of a robust platform for proteome profiling of clinical samples with an unprecedented depth. In fact, the MS ability to generate different levels of information about the individual proteome may lead to the comprehensive characterization of the biological network of pathways involved in SCZ, seeking the identification of reliable biomarkers of the disorder to improve prediction and diagnosis towards the ultimate goal of improving patient care and outcome. 

However, a standardization of the studies’ characteristics is required for more specific clinical proteomics studies. In fact, a precise definition of the study’s objectives and standardization of sociodemographic, clinical, and cognitive variables across the studied groups would make them more objective and specific, allowing a more comprehensive understanding of SCZ pathophysiology and increasing the possibility of identifying specific biomarkers of SCZ. This will minimize the confounding factors, leading to improvements in the statistical power and, consequently, the efficiency of translating biomarker candidates and drug targets to the clinical application associated with the disorder.

The use of MS proteomics pipelines combining (i) standardized conditions; (ii) high-throughput sample preparation techniques; (iii) high computational power for data processing and analysis will lead to a rapid expansion of clinical cohort sizes and consequently to more robust studies. An extra effort should be made to provide data in an open format so the community can re-analyze and perform larger studies based on data analysis from multiple centers. After full implementation of those proteomics pipelines, their application in extended clinical cohorts will allow taking into account the different variables (such as gender, comorbidities, illness duration, and treatment), leading to a more comprehensive understanding of SCZ pathophysiology and, consequently, increasing the possibility of identifying specific biomarkers of SCZ, seeking to improve prediction and diagnosis towards the ultimate goal of improving patient care and outcome.

## 8. Conclusions

Our results highlight the potential of MS-based proteomics strategies to support clinical decisions in the future search for biomarkers for SCZ and the definition of proteome profiles associated with the disorder. The main biological pathways shown as enriched through GO analysis included lipid metabolism, complement and coagulation cascades, and immune response-related pathways, among others. A meta-analysis between schizophrenia patients and healthy controls was performed, with the results suggesting four apolipoproteins, APOA1, APOA2, APOC1 and APOC3 (downregulated), and ficolin-3 (upregulated) as potential biomarkers of the disorder. These proteins should be further studied in larger cohorts to evaluate their potential for disease diagnosis starting with MS-based approaches and moving to more accessible approaches, for example, ELISA assays. Furthermore, post-mortem brain tissue analysis could also include the analysis of these proteins to identify if their peripheral changes reflect their modulation in the brain or a response from a different region of the organism.

This systematic review also highlights several factors that can contribute to the heterogeneity of the findings, including differences in sample size and characteristics, lack of information about illness duration, peripheral fluid sample preparation, analytical methods, and data analysis pipeline. The low number of studies employing validation cohorts and the lack of standardized procedures in reporting data analysis are also important pitfalls of previous studies. Further studies with larger cohorts and validation cohorts, longitudinal designs with multiple collection time-point throughout the evolution of the disorder, the integration of proteomics results with other omics data (phenomics, genomics, metabolomics, connectomics) could provide additional information about differentially expressed proteins in selected biological pathways. The comprehensive information produced could harbor proteomic biomarkers of SCZ, contributing to individualized prognosis and stratification strategies, besides aiding in the differential diagnosis.

## Figures and Tables

**Figure 1 ijms-23-04917-f001:**
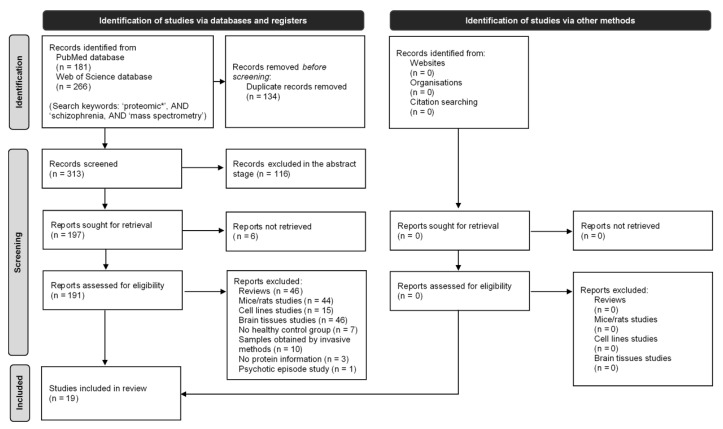
Flow chart of the selection process of the studies included in this systematic review of peripheral fluids MS-based proteomics in SCZ disorder, following PRISMA 2020 [54].

**Figure 2 ijms-23-04917-f002:**
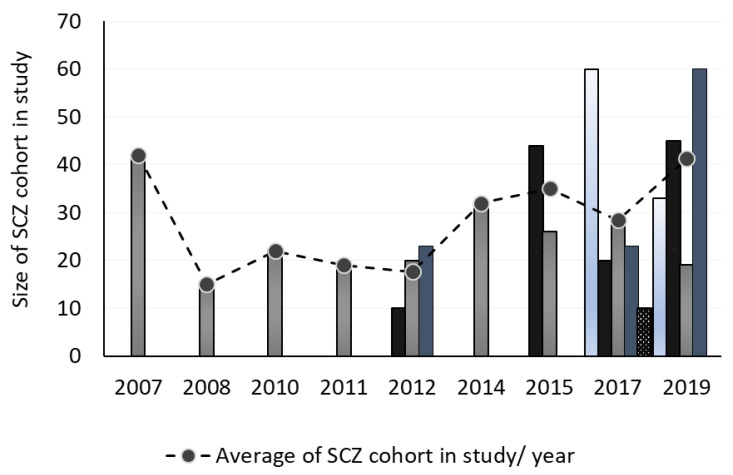
Publication frequency. The number of bars shown in the graphic reflects the number of articles published per year, and the height of each bar reflects the number of SCZ patients in the cohort of the study. The average number of SCZ patients in the cohorts per year is shown in the markers connected by the dashed line.

**Figure 3 ijms-23-04917-f003:**
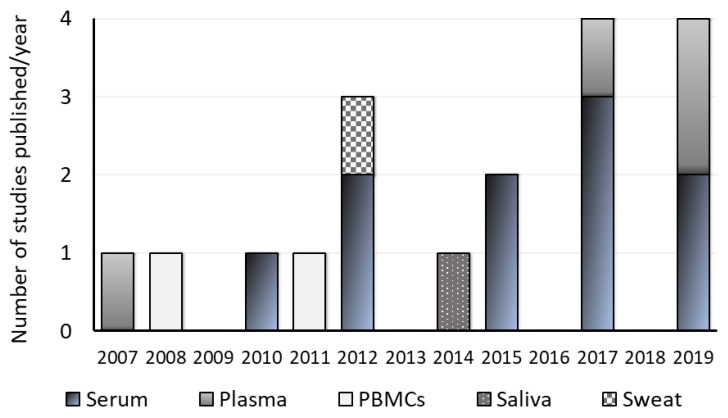
Sample type. The image shows the number of publications per year that fit the criteria of this review. Each color shows the type of samples used, and its height indicates the number of studies.

**Figure 4 ijms-23-04917-f004:**
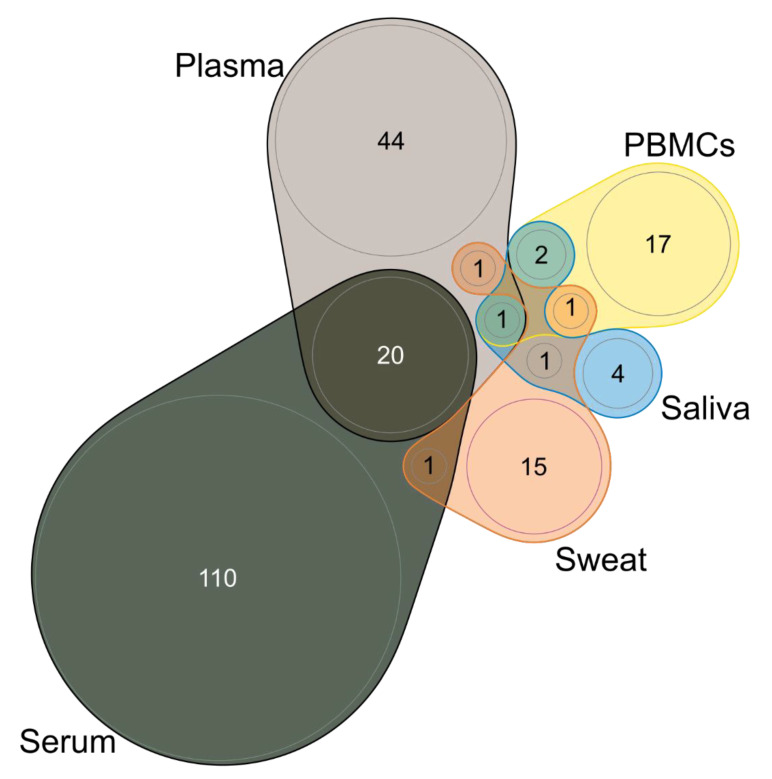
Venn diagram of the 217 proteins identified as altered in the human peripheral fluids serum, plasma, PBMCs, saliva and sweat in the selected studies of schizophrenia (SCZ) vs. control. The proteins identified as altered in: (i) only serum: 110 proteins; (ii) only plasma: 44 proteins; (iii) only PBMCs: 17 proteins; (iv) only saliva: 4 proteins; (v) only sweat: 15 proteins; (vi) plasma vs. PBMCs vs. saliva: 1 protein; (vii) plasma vs. serum: 20 proteins; (viii) plasma vs. PBMCs: 1 protein; (ix) plasma vs. sweat: 1 protein; (x) serum vs. sweat: 1 protein; (xi) PBMCs vs. sweat: 1 protein; (xii) PBMCs vs. saliva: 1 protein; (xiii) sweat vs. saliva: 1 protein.

**Figure 5 ijms-23-04917-f005:**
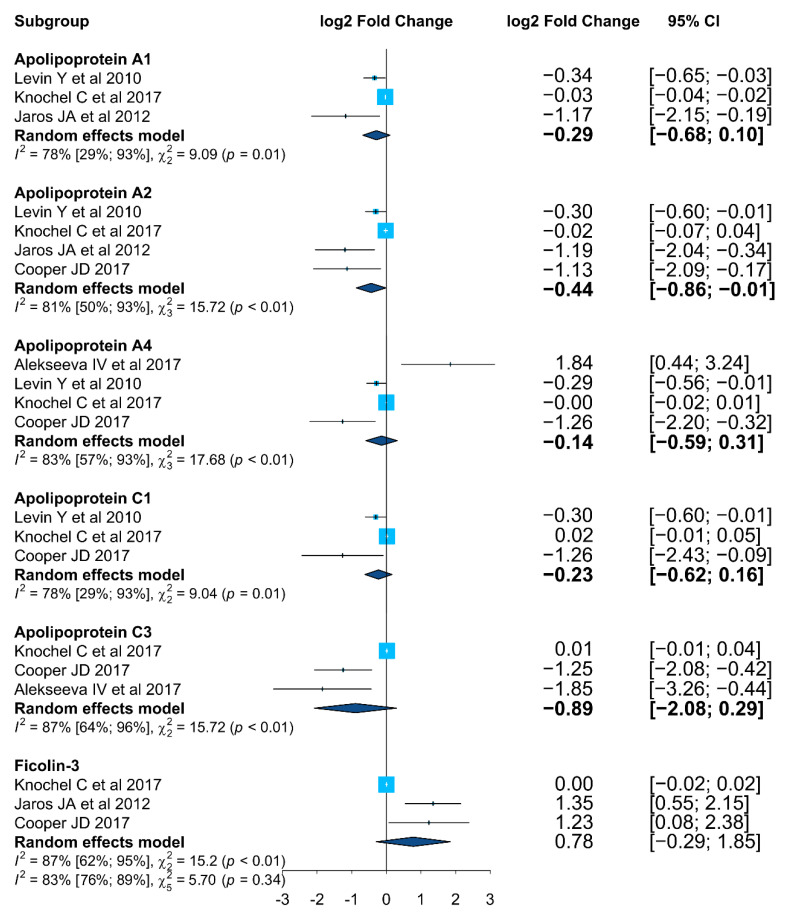
Forest plot from the meta-analysis of proteins identified as altered in SCZ vs. control studies in at least two studies (95% CI, confidence intervals). Squares (whiskers represent 95% CI) indicate the effect sizes of the individual studies. The size of the squares reflects the sample size of each individual study. Diamonds represent summary statistics.

**Figure 6 ijms-23-04917-f006:**
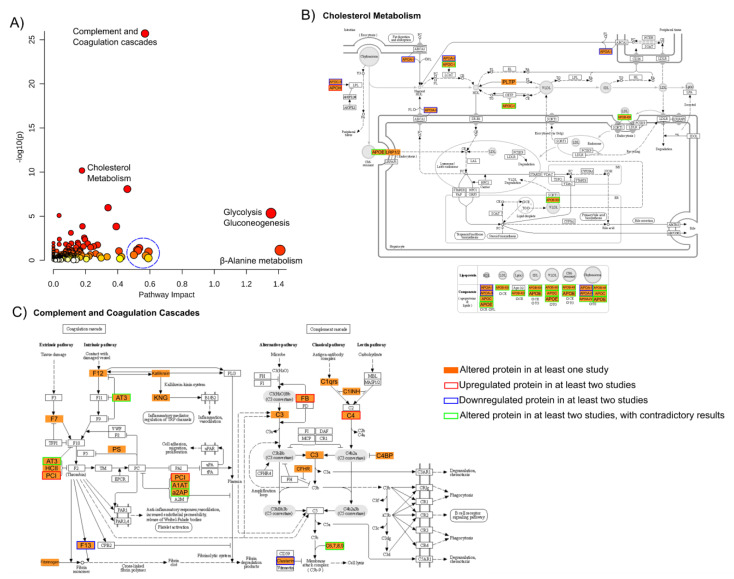
Gene ontology analysis of all proteins considered altered throughout the analyzed reports. A gene ontology approach was used to assess pathway impact and enrichment (here presented by the *p*-value and color scheme in (**A**)) of all proteins described as altered between controls and SCZ in at least one study (Appendix A), represented here as a scatter plot [60]. The blue circle highlights a cluster of ontologies, all belonging to metabolic pathways. From the pathways shown as enriched by this list of proteins, two were selected and their visual representation was obtained through the KEGG Mapper Color tool [61,62]: (**B**) cholesterol metabolism and (**C**) complement and coagulation cascades. In these KEGG panels, the proteins found in any of the studies are shown in orange, and proteins found to be altered in at least two studies are highlighted in red or blue when the protein is always found to be up- or down-regulated in SCZ cases (respectively) or highlighted in green when the results from the two or more studies are contradictory.

**Table 1 ijms-23-04917-t001:** Demographic summary of all the studies included in the systematic review of Schizophrenia and biomarkers discovery using MS-based method in human peripheral fluids.

First Author	Year	Schizophrenia (SCZ)				Controls (CTR)			Other Disorders (OD)				Clinical Criteria	Ref.
		n	Age	Illness Duration	Gender (m/f)	n	Age	Gender (m/f)	n	Age	Illness Duration	Gender (m/f)		
L. Smirnova	2019	33	34 (28–40)	7 (4–16)	11/22	24	28 (21–55)	6/18	23 (BD)	32 (21–52)	8 (5–11)	14/9	ICD-10	[63]
Rodrigues-Amorim	2019	45	41 ± 15	12 ± 11	28/17	43	44 ± 14	26/17	---	---	---	---	DSM-V	[64]
G.S. Pessoa	2019	19	37 ± 11	7.6 ± 5.4	13/6	13	38 ± 16	3/10	19 (BD)	41 ± 17	6.4 ± 6.1	7/12	ICD-10	[65]
C. Walss-Bass	2019	60	43 ± 1.4	---	46/14	20	41 ± 2.6	14/6	---	---	---	---	DSM-IV	[66]
J. D. Cooper	2017	60	31 ± 10	---	31/29	76	32 ± 9.0	43/36	---	---	---	---	ICD-9 and ICD-10	[67]
T. L. Huang	2017	20	38 ± 11	---	9/11	20	39 ± 6.5	7/13	---	---	---	---	DSM-IV	[68]
C. Knochel	2017	29	37 ± 11	12 ± 7.8	21/8	93	34 ± 11	44/39	25 (BD)	38 ± 10	8.9 ± 5.5	19/6	DSM-IV	[69]
J.R. De Jesus	2017	23	34 ± 9	8.7 ± 7.5	17/6	12 (3 HCF; 9 HCNF)	39 ± 9 (HCF); 35 ± 8 (HCNF)	1/2 (HCF); 2/7 (HCNF)	14 (BD); 4 (OD)	36 ± 9 (BD); 31 ± 5 (OD)	4.5 ± 4.3 (BD); 4.5 ± 2.9 (OD)	5/9 (BD); 3/1 (OD)	ICD-10	[70]
I. V. Alekseeva	2017	10	35 ± 13	---	6/4	10	39 ± 11	3/7		---	---	---	ICD-10	[71]
Y. H. Ding	2015	44	33 ± 8.4	---	20/24	40	34 ± 9.2	18/22	26 (DP)	33 ± 8.6	---	11/15	ICD-10	[72]
K. Al Awam	2015	26	37 ± 12	12 ± 12	20/6	26	37 ± 11	20/6	---	---	---	---	DSM-IV	[73]
J. Iavarone	2014	32	---	---	---	31	---	---	17 (BD)	---	---	---	DSM-IV and ICD-10	[74]
Y. Li	2012	10	52 ± 6.4	---	5/5	10	53 ± 6.2	5/5	---	---	---	---	DSM-IV	[75]
J. Jaros	2012	20	31 ± 9.4	---	10/10	20	32 ± 9.3	10/10	---	---	---	---	ICD-10	[76]
M. M. Raiszadeh	2012	8	16 ± 9.7	---	6/2	4	22	---	---	---	---	---	DSM-IV	[77]
M. Herberth	2011	19	30 ± 8.9	---	14/5	19	35 ± 7.2	12/7	---	---	---	---	DSM-IV	[78]
Y. Levin	2010	22	29 ± 11	---	15/7	33	28 ± 7.0	18/15	---	---	---	---	DSM-IV	[79]
R. M. Craddock	2008	15	36 ± 15	---	11/4	15	34 ± 9.6	11/4	---	---	---	---	DSM-IV	[80]
C. Wan	2007	42	34 ± 20	---	26/16	46	39 ± 12	22/24	---	---	---	---	DSM-III	[81]

SCZ: schizophrenia; CTR: control; BD: bipolar disorder; DP: depression; OD: other disorders; HCF: familiar healthy control; HCNF: non-familiar healthy control.

**Table 2 ijms-23-04917-t002:** Proteomic studies of schizophrenia and biomarkers discovery using MS-based method in human peripheral fluids. The proteins identified as altered are represented by their entry name as described in UniProt (the corresponding protein name and accession number are described in Appendix A).

Author (Year)	Cohort Information	Sample	Type of Sampling	Drug Naive	MS-Based Method	Other Techniques	Quantification Method	Depletion/Enrichment	Altered Proteins	Altered Pathways	Ref.
Smirnova (2019)	33 SCZ; 23 BD; 24 CT	Serum	Individual	Yes	1DE-LC-MS/MS	ELISA (Q6UB98; P33151)	MS	Yes/No	**SCZ vs. CTR vs. BD:****↑** (A2ML1; ZN189; SMC2; FA12; AACT; APOE; A2GL; IPSP; DMD; CPN2; ABL2; ACTB; ACTG; PRKDC; DCD; RL19; LRP2; LG3BP; ITSN1; ECM1; ARMX4; ANR12; DHX29; DYH5; PINX1; CNDP1; FETUB); **↓** (TNRC18; APOM; CASB; C1QA; RET4; APOD; TETN; CO8G; CO6; DESP; VGFR1; EST1; CADH5; KI67; MYT1; HORN; MAGE1; GULP1)	**SCZ**: immune response, cell communication, cell growth and maintenance, protein metabolism, and regulation of nucleic acid metabolism. **BD**: immune response, regulating transport processes across the cell membrane and cell communication, development of neurons and oligodendrocytes, and cell growth.	[63]
Rodrigues-Amorim (2019)	45 SCZ (10 FEP; 35 chronic); 43 CT	Plasma	Individual	No	1DE-LC-MS/MS	WB (drebrin, GMFB, BDNF, RAB3GAP1, attractin)	MS	No/Yes	1302 proteins screened and 34 selected (specific funccctions at CNS level). 5 proteins analyzed. **SCZ vs. CT:** **↓** (BDNF; GMFB; RB3GAP1)	Psychoneuroimmune signaling. The available evidence suggests that SCZ causes dysfunction in synaptic, neurotransmission, and neuronal patterns.	[64]
Pessoa (2019)	19 SCZ; 19 BD; 13 CT	Serum	Pooled	No	LC-MS/MS and LC/ICP-MS	---	MS	No/No	**SCZ vs. CT:****↑** (IGHG1; KV320); **↓** (IGKC; IGLC2; TRFE; J3QRN2; IGHG3; KVD28; S4R460; LV325; IGHG2)	Imbalance in the homeostasis of important micronutrients.	[65]
Walss-Bass (2019)	60 SCZ; 20 CT	Plasma	Pooled	No	1DE-LC-MS/MS	ELISA (C4A; APOB)	MS	Yes/Yes	Total ID: 10. **SCZ vs. CT:** **↑** (C4; APOB)	C4 levels in patients are likely due to the presence of the illness itself, while APOB may be a marker of antipsychotic-induced alterations.	[66]
Cooper (2017)	60 SCZ; 77 CT (Cologne study)	Serum	Individual	Yes	LC-MS/MS (MRM mode)	---	MS	No/No	77 proteins (68 analyzed after QC) were quantified of a total of 101 selected proteins. **SCZ vs. CT:** **↑** (HPT; ICI; ANT3; CO4A; AACT; ITIH4; CO9; FCN3; A2AP;APOH); **↓** (APOA2; APOC3; APOA4; APOC1)	Coagulation, metabolism, and inflammation pathways. Suggest that an increased oxidative stress response may represent an inherent SCZ vulnerability.	[67]
Huang (2017)	20 SCZ; 20 CT	PBMCs	Individual	No	MALDI-TOF MS	---	MS	No/No	**SCZ vs. CT:****↑** (Alpha defensins)	Suggested the activation of immune pathway of PBMCs.	[68]
Knochel (2017)	29 SCZ; 25 BD; 93 CT	Plasma	Individual	No	LC-MS/MS (MRM mode)	MRI	MS	No/No	**SCZ vs. CT:****↑** (APOC1, APOC2, APOC3, APOC4, CFAB, CO3, FCN3, KLKB1, MMP9, PEDF); **↓** (A2AP, ANT3, APOA1, APOA2, APOA4, APOB, APOD, APOE, APOF, APOL1, C1QC, F13B, HEP2, HRG, RET4) **SCZ vs. BD:** **↑** (APOC2; APOC4; C1QC; CO3; F13B; KLKB1; MMP9); **↓** (A2AP; ANT3; APOA1; APOA2; APOA4; APOB; APOC1; APOC3; APOD; APOE; APOF; APOL1; CFAB; FCN3; HEP2; HRG; PEDF; RET4)	Altered APOC expression in SCZ and BD was linked to cognitive decline and underlying morphological changes in both disorders.	[69]
De Jesus (2017)	23 SCZ; 14 BD; 4 OD; 12 CT (3 HCF; 9 HCNF)	Serum	Pooled	No	LC-MS/MS	---	2D DIGE	Yes/No	**SCZ vs. BD:****↑** (C4A; C4B; SAMP)	Altered proteins are associated with an inflammatory response.	[70]
Alekseeva (2017)	10 SCZ; 10 CT	Serum	Individual	No	2DE MALDI-TOF/TOF	---	2DE	Yes/No	**SCZ vs. CT:****↑** (APOA4; HPT); **↓** (APOC2; APOC3; SAA1; CLUS; TTHY; ALBU; A1AT; Haptoglobin hp2α (protein ID))	Altered proteins are associated to lipid homeostasis deregulation, and inflammatory response	[71]
Ding (2015)	44 SCZ; 26 DP; 40 CT	Serum	Individual	No	SELDI-TOF-MS and MALDI-TOF MS	---	MS	No/Yes	**SCZ:****↓** (N-terminal fragment of fibrinogen)	---	[72]
Al Awam (2015)	26 SCZ; 26 CT	Serum	Individual	No	MALDI-TOF-MS	GC-MS, FTIR	MS	No/Yes	Total Detected: 94; Significantly different: 11 protein ions. **SCZ:** **↓** (suggested to be a fragment of APOA1)	---	[73]
Iavarone (2014)	32 SCZ; 17 BD; 31 CT	Saliva	Individual	No	LC-MS/MS	---	MS	No/No	**SCZ vs. CT:****↑** (α-defensins 1–4, S100A12, cystatin A and S-derivatives of cystatin B)	SCZ-associated dysregulation of the immune pathway of peripheral white blood cells. Suggested that the dysregulation of the BD group could involve the activation of a more specific cell type than that of the SCZ group.	[74]
Li (2012)	10 SCZ; 10 CT	Serum	Individual	Yes	LC-MS/MS	ELISA	MS	Yes/No	Total ID: 1344. **SCZ vs. CT:** **↑** (CO8B; CD5L; DOPO; IGHG4; IGHM; KNG1; PI16; PGRP2; ITIH4; PLTP; IPSP; IGK@ protein; IGL@ protein); **↓** (AMPN; APOC2; APOF; C4BPB; APOL1; FA7; GGH; ICAM2; ALS; isoforms 2 of ITIH4; LBP; PROS; ZNF57)	Dysregulation of the alternative complement pathway in SCZ patients.	[75]
Jaros (2012)	20 SCZ; 20 CT	Serum	Individual	Yes	LC-MS/MS	ELISA (RET4; FCN3)	MS	Yes/Yes ⁑	Total ID: 312. Significantly different: 35. Phospho altered: 72. **SCZ vs. CT:** **↑** (K2C6B; FCN3; SRBS1; NUCB1; K1C9; NUDT6; ALS2; IBP3; MAST1; CFAB; C4BPA; FHR3; ITIH3; CO6; AGRE1); **↓** (CAH1; RET4; LRRC7; FR1L6; KI21B; TETN; KIF27; APOA1; APOA2; MYOF; FIBA; CCD57; SMC1A; K1C14; PHLD; LIFR; XIRP1 ↓; WDR19; SMC4; SAGE1)	Acute phase; Complement and coagulation system; Immune Response.	[76]
Raiszadeh (2012)	23 SCZ; 55 CT For analysis: 4 SCZ; 4 CT (2nd pool)	Sweat	Pooled	No	LC-MS/MS and LC-MS/MS-MRM	---	MS	No/No	1st set Total ID: 150; 2nd set Total ID: 185; MRM: 30. **SCZ vs. CT:** **↑** (ZA2G; ANXA5; ARG2; BLMH; CALL5; CASPE; CDSN; CSTA; DCD; Desmoglein; DJ-;G3PDH; KLK11; KRT10; PRDX1; PEBP1; S100A7; THIO); **↓** (PIP)	Metabolic process.	[77]
Herberth (2011)	19 SCZ; 19 CT	PBMCs	Individual	Drug naïve/ treated	LC-MS/MS	WB (ALDOC, GAPDH, LDHB, PGK1, TPIS)	MS	No/Yes	**Unstimulated PBMCs:****↑** (CNDP2; Uncharacterized protein KIAA0423; LDHB); **↓** (COTL1; GPI; HSP72). **Stimulated PBMCs:** **↑** (ALDOC; GAPDH; HNRPK; LDHB; MYH14; MYH15; NAMPT; PGK1; PPIA; TPIS; PKLR; PGAMA4); **↓** (CH60).	Glycolytic pathway, Immune response.	[78]
Levin (2010)	22 SCZ; 33 CT	Serum	Individual	No	LC-MS/MS	ELISA (APOA1; APOA2; APOA4; FETUA)	MS	Yes/No	Total ID: 1411. Significantly different: 10. **SCZ vs. CT:** **↑** (CD5L; IGHM; F13B; TRFE; APOD; APOA1; FETUA; APOA4; APOA2; APOC1)	Lipid metabolism; molecular transport; Immune response.	[79]
Craddock (2008)	15 SCZ; 15 CT	PBMCs	Individual	Yes	SELDI-TOF-MS	ELISA (α-defensins)	MS	No/Yes	**SCZ:****↑** (α-defensins)	Immune alteration.	[80]
Wan (2007)	42 SCZ; 46 CT	Plasma	Individual	No	MALDI-TOF MS	---	2-DE	No/No	**SCZ vs. CT:****↑** (Haptoglobin a; a1-Antitrypsin; a1-Microglobulin; SAMP; ANT3; VTDB);	Evidence indicates that chronic systemic inflammation may be an aetiological agent of the pathophysiology of SCZ.	[81]

BD: bipolar disorder; CNS: central nervous system; CT: controls; DP: depression; ELISA: enzyme-linked immunosorbent assay; FEP: first-episode psychosis; HCF: familiar healthy control; HCNF: non-familiar healthy control; OD: other disorders; SCZ: schizophrenia; WB: Western blot; ⁑ Despite the enrichment method used, the flow-through was also analyzed.

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
