# Peer review of "Systematic Review and Meta-Analysis of Mass Spectrometry Proteomics Applied to Human Peripheral Fluids to Assess Potential Biomarkers of Schizophrenia"

_ijms, 2022, doi:10.3390/ijms23094917_

Round 1
Reviewer 1 Report
This is a well written report and since it will be open access, it will be a good source of information for the respective community. However, the manuscript needs some careful proofreading for grammar and missed out lines. For instance, the author list is incomplete.
Please complete 'and 8'. There is no author name provided there.
Author Response
We acknowledge the reviewer’s comments to our work. We made changes to some typos and corrected some of the formatting, including the author issue.
Reviewer 2 Report
The present systematic review and meta-analysis explores the role of mass spectrometry in human peripheral fluids and the proteomic approach to enhance diagnostic accuracy in Schizophrenia.
The systematic review included a total of 19 studies, with meta-analytic investigations conducted on concentrations of six proteins.
The study is timely and very interesting, and the explored topic is of great scientific and clinical value. The manuscript is well-written and easy to read, figures are clear and informative .
However, some important methodological issues have to be addressed in order for the paper to represent a valuable contribution to the research field.
MAJOR ISSUES
-The systematic literature search per the present study has been conducted considering 2 databases, PubMed (Embase) and Web Of Science. While these are certainly two of the most important databases, it is recommended to search at least 3 relevant databases (see Bramer MW et al. Optimal database combinations for literature searches in systematic reviews: a prospective exploratory study. Syst Rev 6, 245 (2017). https://doi.org/10.1186/s13643-017-0644-y and Higgins J et al., Cochrane Handbook for Systematic Reviews of Interventions Version 6.3, 2022). Given the nature on research, it would be advisable to conduct a further search on one more database (considering the topic, Scopus or PsycInfo could also represent valuable databases for the search).
-The search string should be better specified. Were the keywords searched in the full text, in title and abstract, or only in studies keywords? In the latter case, the search can not be considered truly systematic and should be performed again.
- Regarding meta-analytical findings, results are inappropriately interpreted. It manuscript it is written tha “APOA1, APOA2, APOC1, and APOC3 show a consistent decrease in SCZ patients compared with healthy control subjects” (lines 556 and 557). However, for APOA1, APOC1, and APOC3 the 95% CI include 0, suggesting that no significant difference can be observed between subjects and health controls. This is reflected in the figure, where the diamond clearly crosses the no-significance line.
MINOR ISSUES
-In the introduction, it is stated: “Epidemiologic studies show that it can take up to several years between symptom onset and diagnosis; evidence suggests that the earlier the diagnosis, the better the prognosis, by decreasing the duration of untreated psychosis.” (lines 76 and 77). While this is certainly correct, more references to support this statement should be provided.
-Also in the Introduction, the Authors report that “While positive symptoms can stabilize throughout the course of the illness, negative symptoms tend to increase and become 82 chronic along with cognitive impairments.” (lines 81 and 82). While this is accurate in untreated individuals, treatment that can successfully reduce negative symptoms (see Galderisi S et al., EPA guidance on treatment of negative symptoms in schizophrenia. Eur Psychiatry. 2021;64(1):e21 doi:10.1192/j.eurpsy.2021.13 ) as well as cognitive impairment (see Vita A et al., Effectiveness, Core Elements, and Moderators of Response of Cognitive Remediation for Schizophrenia: A Systematic Review and Meta-analysis of Randomized Clinical Trials. JAMA Psychiatry. 2021;78(8):848-858. doi:10.1001/jamapsychiatry.2021.0620) in schizophrenia have been developed and are currently being implemented in clinical practice. This should be explicitly reported in the introduction for better clarity for the reader.
Author Response
Reviewers comments followed by our answers in bullet points
The present systematic review and meta-analysis explores the role of mass spectrometry in human peripheral fluids and the proteomic approach to enhance diagnostic accuracy in Schizophrenia.
The systematic review included a total of 19 studies, with meta-analytic investigations conducted on concentrations of six proteins.
The study is timely and very interesting, and the explored topic is of great scientific and clinical value. The manuscript is well-written and easy to read, figures are clear and informative .
However, some important methodological issues have to be addressed in order for the paper to represent a valuable contribution to the research field.
MAJOR ISSUES
-The systematic literature search per the present study has been conducted considering 2 databases, PubMed (Embase) and Web Of Science. While these are certainly two of the most important databases, it is recommended to search at least 3 relevant databases (see Bramer MW et al. Optimal database combinations for literature searches in systematic reviews: a prospective exploratory study. Syst Rev 6, 245 (2017). https://doi.org/10.1186/s13643-017-0644-y and Higgins J et al., Cochrane Handbook for Systematic Reviews of Interventions Version 6.3, 2022). Given the nature on research, it would be advisable to conduct a further search on one more database (considering the topic, Scopus or PsycInfo could also represent valuable databases for the search).
- we appreciate the comment and the indication of the recommendation of the 3 databases. Using 2 databases is the minimum to perform a systematic review and a meta-analysis according to the PRISMA Statment. This recommendation of using 3 databases would lead to a considerable increase in the workload and significant delays in this publication. Also, we used two of the largest and most used databases (NCBI, WebOfScience and some manual documents were also added, as reported)
-The search string should be better specified. Were the keywords searched in the full text, in title and abstract, or only in studies keywords? In the latter case, the search can not be considered truly systematic and should be performed again.
- the search was not limited to any field. This information was added to the manuscript. “The search was performed in all fields”
- Regarding meta-analytical findings, results are inappropriately interpreted. It manuscript it is written tha “APOA1, APOA2, APOC1, and APOC3 show a consistent decrease in SCZ patients compared with healthy control subjects” (lines 556 and 557). However, for APOA1, APOC1, and APOC3 the 95% CI include 0, suggesting that no significant difference can be observed between subjects and health controls. This is reflected in the figure, where the diamond clearly crosses the no-significance line.
- we agree with the reviewer that our initial text was not in accordance with the data. We changed the text accordingly. Most studies show a decrease but in fact the meta-analysis does not show a “consistent decrease”.
MINOR ISSUES
-In the introduction, it is stated: “Epidemiologic studies show that it can take up to several years between symptom onset and diagnosis; evidence suggests that the earlier the diagnosis, the better the prognosis, by decreasing the duration of untreated psychosis.” (lines 76 and 77). While this is certainly correct, more references to support this statement should be provided.
- à more references were added to support this sentence.
-Also in the Introduction, the Authors report that “While positive symptoms can stabilize throughout the course of the illness, negative symptoms tend to increase and become 82 chronic along with cognitive impairments.” (lines 81 and 82). While this is accurate in untreated individuals, treatment that can successfully reduce negative symptoms (see Galderisi S et al., EPA guidance on treatment of negative symptoms in schizophrenia. Eur Psychiatry. 2021;64(1):e21 doi:10.1192/j.eurpsy.2021.13 ) as well as cognitive impairment (see Vita A et al., Effectiveness, Core Elements, and Moderators of Response of Cognitive Remediation for Schizophrenia: A Systematic Review and Meta-analysis of Randomized Clinical Trials. JAMA Psychiatry. 2021;78(8):848-858. doi:10.1001/jamapsychiatry.2021.0620) in schizophrenia have been developed and are currently being implemented in clinical practice. This should be explicitly reported in the introduction for better clarity for the reader.
- This issue was now included in the introduction.
Reviewer 3 Report
This is an interesting systematic review and meta-analysis evaluating the efficacy of mass spectrometry techniques applied to human peripheral fluids in patients with schizophrenia. The authors aimed to explore disease biomarkers and relevant biological pathways implicated in the field of schizophrenia. The paper is well-written, and interesting for the readers. However, several minor changes should be made before being considered for publication.
The introduction section is mainly focused, firstly on the biological basis of neuropsychiatric disorders, schizophrenia, the search for biomarkers and the biomarkers in psychiatric disorders.
I would recommend to add some more references from experts in the field. For instance there is an important paper on "the search for new biomarkers for cognition in schizophrenia" by Penadés that reviewed the relationship between inflammation and cognition, the role of prolaction and BDNF. Several lines summarizing it, would improve the introduction.
At the end of the introduction section, the authors should provide an expanded explanation of the aims of the study.
The methods section should be started by describing that they carried out a systematic review based on two databases. In a second step, they shoudl report hwo they did the review.
Figure 1 and overall methods are based on the PRISMA 2020. The quality of the methods and description of them is really good.
Table II should be renamed as Table 2. at the first column the authors should report the first author (et al.) and the year of publication.
Sections and subsection should be renumbered.
The results are well reported.
In the conclusions section, the authors shoudl report the main findings and how these findings are enriching the literature in the field. They have included some lines about further studies to be carried out. However, they should also propose a potential line of research for future studies including all of these potential biomarkers.
Author Response
Reviewer comments followed by our answers in bullet points.
This is an interesting systematic review and meta-analysis evaluating the efficacy of mass spectrometry techniques applied to human peripheral fluids in patients with schizophrenia. The authors aimed to explore disease biomarkers and relevant biological pathways implicated in the field of schizophrenia. The paper is well-written, and interesting for the readers. However, several minor changes should be made before being considered for publication.
The introduction section is mainly focused, firstly on the biological basis of neuropsychiatric disorders, schizophrenia, the search for biomarkers and the biomarkers in psychiatric disorders.
I would recommend to add some more references from experts in the field. For instance there is an important paper on "the search for new biomarkers for cognition in schizophrenia" by Penadés that reviewed the relationship between inflammation and cognition, the role of prolaction and BDNF. Several lines summarizing it, would improve the introduction.
- we appreciate the recommendation, and this work was added in the document (in the GO analysis where inflammation is discussed)
At the end of the introduction section, the authors should provide an expanded explanation of the aims of the study.
- We have reformulated this section accordingly.
The methods section should be started by describing that they carried out a systematic review based on two databases. In a second step, they shoudl report hwo they did the review.
- This section was updated with this information in the section 2.1, which followed the PRISMA protocol and the PROSPERO identifier.
Figure 1 and overall methods are based on the PRISMA 2020. The quality of the methods and description of them is really good.
Table II should be renamed as Table 2. at the first column the authors should report the first author (et al.) and the year of publication.
- The document was changed accordingly
Sections and subsection should be renumbered.
- the document was changed accordingly
The results are well reported.
In the conclusions section, the authors shoudl report the main findings and how these findings are enriching the literature in the field. They have included some lines about further studies to be carried out. However, they should also propose a potential line of research for future studies including all of these potential biomarkers.
- a brief description was added to this section considering the importance of the reviewer comments.
Round 2
Reviewer 2 Report
The Authors have responded in a satisfactory manner to all queries and the manuscript is consistently improved.